# Comparison of ZnO, Al_2_O_3_, AlZnO, and Al_2_O_3_-Doped ZnO Sensing Membrane Applied in Electrolyte-Insulator-Semiconductor Structures

**DOI:** 10.3390/membranes12020168

**Published:** 2022-01-30

**Authors:** Chyuan-Haur Kao, Yi-Wen Liu, Chih-Chen Kuo, Shih-Ming Chan, Deng-Yi Wang, Ya-Hsuan Lin, Ming-Ling Lee, Hsiang Chen

**Affiliations:** 1Department of Electronic Engineering, Chang Gung University, 259 Wen-Hwa 1st Road, Kwei-Shan District, Taoyuan City 333, Taiwan; chkao@mail.cgu.edu.tw (C.-H.K.); chkao@mail.cgu.edu (Y.-W.L.); 2Kidney Research Center, Department of Nephrology, Chang Gung Memorial Hospital, Chang Gung University, No. 5, Fuxing St., Guishan Township, Taoyuan City 333, Taiwan; 3Department of Electronic Engineering, Ming Chi University of Technology, 284 Gungjuan Rd., Taishan Dist., New Taipei City 243, Taiwan; 4Department of Applied Materials and Optoelectronic Engineering, National Chi Nan University, No.1, University Rd., Puli 545, Taiwan; s107328003@mail1.ncnu.edu.tw (C.-C.K.); s107328009@mail1.ncnu.edu.tw (S.-M.C.); s109328505@mail1.ncnu.edu.tw (D.-Y.W.); s108328042@mail1.ncnu.edu.tw (Y.-H.L.); 5Department of Electro-Optical Engineering, Minghsin University of Science and Technology, No. 1, Xinxing Rd., Xinfeng 304, Taiwan

**Keywords:** ZnO, Al_2_O_3_, sensing membranes, annealing, defects, Al concentration

## Abstract

In this study, ZnO, AlZnO, Al_2_O_3_, and Al_2_O_3_-doped ZnO-sensing membranes were fabricated in electrolyte–insulator–semiconductor (EIS) structures. Multiple material analyses indicate that annealing at an appropriate temperature of 500 °C could enhance crystallizations, passivate defects, and facilitate grainizations. Owing to their material properties, both the pH-sensing capability and overall reliability were optimized for these four types of membranes. The results also revealed that higher Al amounts increased the surface roughness values and enhanced larger crystals and grains. Higher Al compositions resulted in higher sensitivity, linearity, and stability in the membrane.

## 1. Introduction

Since the first ISFET was invented by Bergervald in 1968, attention has been drawn to the development of semiconductor-based sensors capable of detecting ion concentrations in solutions [1]. Measurements of ion concentrations in blood, urine, or other body fluids are crucial for monitoring human health conditions [2], and semiconductor-based sensors have the advantages of compact size, rapid response, easy fabrication, and long-term reliability [3,4]. As proposed in the literature [5,6], strategies to integrate various materials into a semiconductor field-effect device are some of the most attractive current trends in the fabrication field. Sensitive field-effect semiconductor devices, such as ion-sensitive field-effect transistors (ISFETs), electrolyte–insulator–semiconductor (EIS) capacitors, and light-addressable potentiometric sensors (LAPS) represent typical semiconductor ion sensors [3]. In an EIS capacitor, a dielectric functioning as a sensing membrane between an electrolyte solution and the semiconductor is crucial for sensing performance. In a conventional EIS device, the dielectric layer is SiO_2_, which has the disadvantages of ineffective electric field crossing, low capacitance, and threshold voltage instability. Recently, various high-k dielectrics, including Ta_2_O_5_, HfO_2_, Y_2_O_3_, and ZrO_2_, have emerged to replace SiO_2_ as the sensing membrane [5,6,7,8]. These dielectrics can have better sensing performance owing to effective electric field crossing in the oxides and high thermal stability. More specifically, the drift can be reduced to very low values in an Al_2_O_3_ dielectric. Furthermore, Al_2_O_3_ has several advantages, including being wear-resistant, having excellent dielectric properties from DC to GHz frequencies, good thermal conductivity, and strong acid and alkali attack resistance [9]. In addition to traditional dielectric materials, transparent conductive oxides (TCO), including SnO_2_ and ZnO, have emerged as sensing membranes for future possible integration with optical semiconductor devices [10,11]. Therefore, ZnO possessing a wurtzite structure, a wide bandgap of 3.37 eV, and a large exciton bandgap of 60 meV has also been utilized for sensing membrane materials [12]. It is, therefore, worth combining Al_2_O_3_ and ZnO to form a multifunctional EIS; however, the influence of Al or Al_2_O_3_ concentrations on ZnO membranes has not been clearly reported yet [13]. In this study, ZnO, Al-doped ZnO, Al_2_O_3_-doped ZnO, and Al_2_O_3_ EIS sensing membranes were compared [14], and multiple material, electrical, and sensing measurements were performed. The results indicate that higher concentrations of Al_2_O_3_ may enhance the sensing capability, owing to the increase in the sensing surface sites and the decrease in the diffusion capacitance in the solution [15]. Our research shows that the addition of Al_2_O_3_ or Al can optimize the sensing behaviors of ZnO sensing membranes.

## 2. Experimental

To apply ZnO, AlZnO, and Al_2_O_3_-doped ZnO and Al_2_O_3_ onto EIS structures, fabrication was performed on a 4-inch n-type (100) silicon wafer with a resistivity of 5–10 Ω-cm. To remove the native oxide, the silicon wafer was cleaned using HF (HF: H_2_O = 1:100). Then, in the first and second conditions, ZnO and AlZnO films 50 nm in thickness were deposited on the silicon wafer by radiofrequency (RF) reactive sputtering with a mixture of Ar and O_2_ (Ar:O_2_ = 20:5) ambient. In the third condition, a 50-nm Al_2_O_3_-doped ZnO sensing membrane was deposited by co-sputtering on an n-type silicon wafer. During the reactive sputtering, ZnO and Al_2_O_3_ targets were used in the ambient of Ar:O_2_ at 20:5. The radio frequency (RF) power was 100 W, and the pressure was 10 mTorr, respectively. Rapid thermal annealing (RTA) was used to anneal the samples at temperatures of 500 °C, 600 °C, and 700 °C in O_2_ ambient for 30 s. An Al film of 300 nm in thickness was then deposited on the backside of the silicon wafer. Next, an epoxy bond was used to define the sensing area. Finally, the samples were fabricated on the copper lines of a printed circuit board (PCB) in silver gel. An epoxy package was used to separate the EIS structure and the copper lines. A detailed EIS structure containing membranes is illustrated in Figure 1.

## 3. Results and Discussion

To investigate the effects of annealing, XRD was used to research the crystalline structure of ZnO, AlZnO, Al_2_O_3_, and Al_2_O_3_-doped ZnO films before and after RTA treatment, as shown in Figure 2. In the figure of the ZnO as-deposited sample, a diffraction crystalline ZnO phase of peak intensity (100) at 33.50° [16]. can be observed. After post-annealing, the peak intensity increased as the temperature rose, and the sample with an RTA temperature of 500 °C clearly exhibited a strong peak of (100). However, for the AlZnO, Al_2_O_3_ and Al_2_O_3_-doped ZnO samples, the films annealed from 500 °C to 700 °C exhibited major peaks at (002), indicative of a polycrystalline structure, as shown in Figure 2b–d. At an RTA temperature of 500 °C, XRD patterns clearly exhibited a strong peak of (002) and the formation of a well-crystallized AlZnO, Al_2_O_3_, and Al_2_O_3_-doped ZnO structure. Further, Figure 2c shows the Al_2_O_3_ as-deposited sample, and the crystalline phase of Al_2_O_3_ can be clearly observed to have characteristic XRD peaks at 46.71°, 61.37°, and 67.94, and diffraction peaks of (400), (511), and (440), respectively [17]. The XRD peaks of AlZnO, Al_2_O_3_, and Al_2_O_3_-doped ZnO film increased in intensity with annealing, which may have been caused by the enhancement of lattice structures and the formation of higher peak intensities. For all four types of samples, the strongest crystallized films could be formed at an annealing temperature at 500 °C, which may be the most preferable annealing condition for the sensing membrane.

In order to monitor the chemical bindings and element compositions, O 1s XPS analysis was performed on the samples, as shown in Figure 3a–c. Figure 3a reveals that RTA at an appropriate temperature of 500 °C could effectively suppress the formation of ZnO silicate and optimize the sensing device performance. However, as the annealing temperature further increased to 600 °C, the amount of silicate increased again [18], and annealing at 500 °C could effectively suppress the formation of AlZn silicate in the XPS spectra, as shown in Figure 3b. Moreover, in the O 1s spectra of Al_2_O_3_, Al-OH, which is related to defects, could be effectively mitigated, as shown in Figure 3c [19]. As shown in Figure 3d, for the Al_2_O_3_-doped ZnO samples, annealing at 500 °C could suppress Al–Zn silicate and strengthen both ZnO and Al_2_O_3_ phases. A high concentration of Zn silicate in the membrane may contain plenty of dangling bonds and traps, which may interfere with the effective electric field across the membrane and worsen the modulation of membrane capacitance to evaluate the electrolyte concentrations. Consistent with the XRD patterns, the optimal annealing condition and most desirable material quality occurred at an annealing temperature of 500 °C. Annealing at 500 °C could effectively decrease the silicate/ZnO concentration ratio compared to the as-deposited sample, as shown in the XPS spectra of Figure 3d. In the XRD patterns of Figure 2d, it is clear that AlZnO has much stronger peaks than the as-deposited sample because annealing could remove the silicate and generate highly crystallized films. Therefore, the trend could be observed both in the XPS spectra and the XRD patterns.

In order to examine the surface roughness of the four samples, AFM images were taken. Figure 4a–d shows AFM images of the ZnO film surface. The results indicate that annealing at 500 °C could cause the roughest surface on the ZnO film. Since grainization may cause better crystallization and less defect-related silicate, ZnO annealed at 500 °C may have the most preferable material properties. Moreover, Figure 4e–h shows AFM images of AlZnO film annealed under different conditions. Similar to the previous images, the results show that annealing at 500 °C could enhance grainization and increase surface roughness values.

Consistent with XRD and XPS analyses, annealing at 500 °C formed the most efficient sensing film. Figure 4i–l shows AFM images of the Al_2_O_3_ film surface after RTA at different temperatures. The results indicate that the sample with RTA annealing at 500 °C also had the roughest surface, indicative of good Al_2_O_3_ grainization. Further, the AFM images in Figure 4m–p show Al_2_O_3_-doped ZnO samples annealed at various temperatures. Compared to the as-deposited sample, annealing at 500 °C effectively increased the surface roughness value. 

However, as the annealing temperature increased to 600 °C and 700° C, the roughness increased drastically. Instead of further increasing the sensing capability, a drastic increase in the film roughness value may have worsened the sensing behavior.

The RMS values of the four samples under various treatments are shown in Figure 5. Based on AFM analysis, the most preferable annealing temperature occurred at 500 °C. Furthermore, when the roughness values of the four different samples annealed at 500 °C were compared, Al_2_O_3_ had the largest surface roughness value, followed by Al_2_O_3_-doped ZnO and AlZnO. The smallest roughness value was from the ZnO sample. Our study indicates that a larger grain size (though not extremely large) may increase the crystal size on the sensing membrane and enhance sensing behavior.

To measure the sensitivity and linearity of EIS capacitors, a Ketheley 2400 Sourcemeter was used to evaluate the C–V curves of the samples. As the reference capacitance was set to 0.4 Cmax, the sensitivity and linearity could be calculated by extracting the change rate of the voltage at 0.4 Cmax versus the pH values. Figure 6a,b shows the C–V curves of ZnO sensing film at different RTA temperatures. The sensitivity values of the above four samples for as-deposited and RTA annealing at 500 °C were 30.75 mV/pH and 43.15 mV/pH, respectively. The linearity values of the two samples for as-deposited and RTA annealing at 500 °C were 97.72% and 98.85%, respectively. The C–V curves of the AlZnO sample without annealing and the sample after annealing at 500 °C are shown in Figure 6c,d. The sensitivity values of the samples were 33.45 mV/pH and 49.62 mV/pH, respectively, and the linearity values of the samples were 96.32% and 98.56%, respectively. These results indicate that the addition of Al can boost the sensitivity of the device. Further, the C–V curves of the as-deposited Al_2_O_3_ sample and the sample after annealing at 500 °C are shown in Figure 6e,f. The sensitivity values of the as-deposited sample and the sample after RTA annealing were 41.09 mV/pH and 55.61 mV/pH, respectively. The linearity values of the above two samples were 98.45% and 99.02%, respectively. Additionally, the C–V curves of the as-deposited Al_2_O_3_-doped ZnO sample and the sample after annealing at 500 °C are shown in Figure 6g,h. The sensitivity values of the as-deposited sample and the sample after RTA annealing at 500 °C were 31.17 mV/pH and 45.83 mV/pH, respectively. The linearity values of the two samples were 98.11% and 98.77%, respectively. To compare all the sensing results, pure Al_2_O_3_ sensing membranes annealed at 500 °C had the highest sensitivity and linearity values. However, for future possible integration of ZnO-related materials, ZnO membranes incorporating Al atoms could also effectively increase the sensitivity and linearity.

To compare the sensing behaviors, the sensitivity and linearity extracted from the C–V curves of the ZnO, AlZnO, Al_2_O_3_, and Al_2_O_3_-doped ZnO sensing membranes are graphed in Figure 7a–d, respectively. The data show that RTA annealing at 500 °C boosted the sensitivity and linearity of all four samples. Moreover, the Al_2_O_3_ membrane had the highest sensitivity, followed by AlZnO and Al_2_O_3_-doped ZnO. This is because aluminum can effectively combine with oxygen to form strong bonds and large grains [20]. The results of the C–V curve measurements are summarized in Figure 7a–d.

To study the hysteresis effects of the membranes, the tested samples were put in buffer solutions of different pH values in an alternating cycle (pH 7, pH 4, pH 7, pH 10, and pH 7) of five minutes for each solution. The samples were subjected to a pH loop of 7→4→7→10→7 over a period of 30 min. Hysteresis voltage is defined as the gate voltage difference between the initial and the terminal voltages. The interior defects of dangling bonds and traps can bind with ions in the solutions and induce hysteresis shift. In addition, because the sizes of H+ and OH− ions are different, the diffusion speed of the H+ ions into the membrane’s surface is faster than that of the OH- ions, as described by Bousse et al.. The hysteresis voltage of the ZnO, AlZnO, Al_2_O_3_, and Al_2_O_3_-doped ZnO samples are shown in Figure 8a–d, respectively. Figure 8a shows that the hysteresis voltage of the ZnO for the as-deposited sample and the samples annealed at 500 °C, 600 °C, and 700 °C were 39.77 mV, 7.43 mV, 10.02 mV, and 29.74 mV, respectively. As shown in Figure 8b, the hysteresis voltage of the AlZnO for the as-deposited sample and the samples annealed at 500 °C, 600 °C, and 700 °C were 18.17 mV, 3.81 mV, 7.82 mV, and 17.49 mV, respectively. Figure 8c shows that the hysteresis voltage of the Al_2_O_3_ for the as-deposited sample and the samples annealed at 500 °C, 600 °C, and 700 °C were 12.03 mV, 1.49 mV, 5.52 mV, and 8.39 mV, respectively. Figure 8d shows that the hysteresis voltage of the Al_2_O_3_-doped ZnO for the as-deposited sample and the samples annealed at 500 °C, 600 °C, and 700 °C were 21.26 mV, 4.07 mV, 9.85 mV, and 19.27 mV, respectively. The results indicate that annealing at 500 °C can reduce the number of dangling bonds and traps, and therefore, suppress the hysteresis voltage of the four types. ZnO membranes without Al atoms had higher hysteresis voltages than the other three types of membranes containing Al atoms. Therefore, the incorporation of Al into ZnO films seems to have a tendency to decrease the hysteresis voltage. To compare the hysteresis voltage among all the membranes, pure Al_2_O_3_ sensing membranes annealed at 500 °C had the lowest hysteresis voltage. However, for the future combination of ZnO and Al_2_O_3_ materials, ZnO membranes incorporating Al atoms could also effectively suppress the hysteresis voltage.

Finally, to investigate the drift rate of the membranes for long-term reliability, the samples were submerged in pH7 buffer solutions for twelve hours. Gate voltage drifts were caused by a trap-limited transport mechanism and dispersive transport.

The voltage–time (V–t) curves of the drift voltages of the ZnO, AlZnO, Al_2_O_3_, and Al_2_O_3_-doped ZnO sensing films are shown in Figure 9a–d, respectively. Figure 9a shows that the drift rate values of ZnO for the as-deposited sample and the samples annealed at 500 °C, 600 °C, and 700 °C were 3.76 mV/hr, 0.89 mV/hr, 0.97 mV/hr, and 1.86 mV/hr, respectively. Figure 9b shows that the drift voltage shifts of AlZnO for the as-deposited sample and the samples annealed at 500 °C, 600 °C, and 700 °C were 2.14 mV/hr, 0.71 mV/hr, 0.74 mV/hr, and 1.61 mV/hr, respectively. Figure 9c shows that the drift voltage shifts of Al_2_O_3_ for the as-deposited sample and the samples annealed at 500 °C, 600 °C, and 700 °C were 1.76 mV/hr, 0.47 mV/hr, 0.57 mV/hr, and 1.02 mV/hr, respectively. Figure 9d shows that the drift voltage shifts of Al_2_O_3_ for the as-deposited sample and the samples annealed at 500 °C, 600 °C, and 700 °C were 12.36 mV/hr, 0.76 mV/hr, 0.81 mV/hr, and 1.71 mV/hr, respectively. Notably, the four samples annealed at 500 °C had the lowest drift rates, and the membranes containing Al had smaller drift voltage shifts than the ZnO sample. This smaller drift rate was attributed to the formation of Al_2_O_3_ between the Al atoms and the oxygen atoms, which increased the binding strength, compensated for oxygen vacancies, and reduced lattice defects. To compare the drift voltage among all the membranes, pure Al_2_O_3_ sensing membranes annealed at 500 °C had the lowest drift voltages. However, to integrate Al_2_O_3_ and ZnO for future optical and optoelectronic applications, ZnO membranes incorporating Al atoms could also effectively suppress the hysteresis voltage.

## 4. Conclusions

In this study, ZnO, AlZnO, Al_2_O_3_, and Al_2_O_3_-doped ZnO sensing membranes were formed by sputtering or co-sputtering. To analyze the material properties, multiple material characterizations were performed. Our results indicate that annealing at an appropriate temperature of 500 °C can enhance crystallized phases, as revealed by XRD, suppress the defect-related bindings, as indicated by XPS, and increase the surface roughness values, as shown by AFM. Moreover, the improvements in material properties can enhance the sensing capability. The resulting analyses indicate that the four types of membranes annealed at 500 °C had the most preferable pH sensing sensitivity, linearity, and stability. Furthermore, the results also show that Al_2_O_3_ had the strongest sensing capability and grainization, followed by Al_2_O_3_-doped ZnO and AlZnO films. Our research indicates that increasing Al compositions in the membranes may improve grainization and sensitivity. Our results show that EIS structures using these four types of membranes show promise for future ion-sensing applications and integration with Al-based and Zn-based electronic and optoelectronic devices.

## Figures and Tables

**Figure 1 membranes-12-00168-f001:**
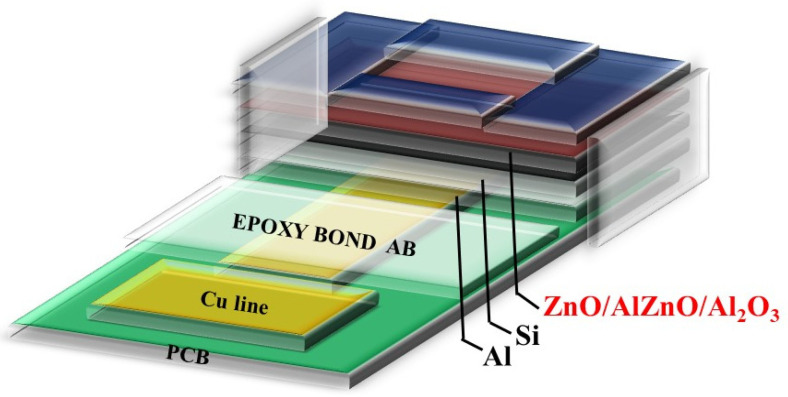
The EIS structure containing membranes.

**Figure 2 membranes-12-00168-f002:**
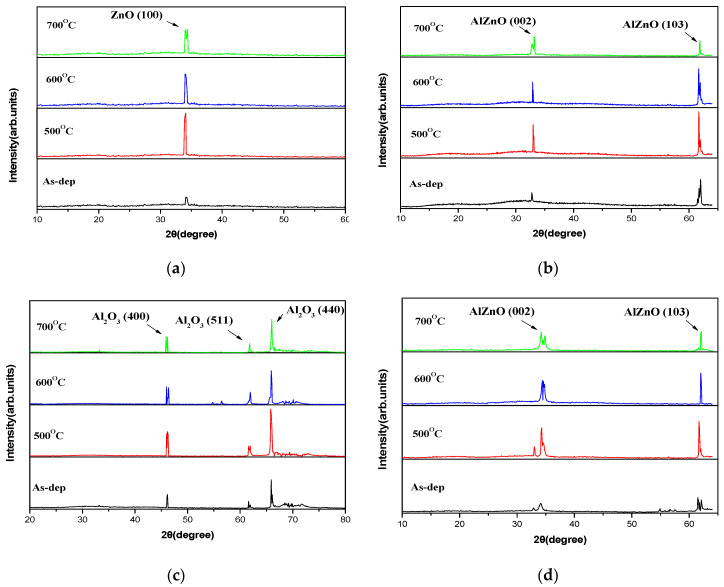
XRD of patterns of (**a**) ZnO, (**b**) AlZnO, (**c**) Al_2_O_3_, and (**d**) Al_2_O_3_-doped ZnO film before and after annealing at various temperatures on single crystalline silicon in O_2_ ambient for 30 s.

**Figure 3 membranes-12-00168-f003:**
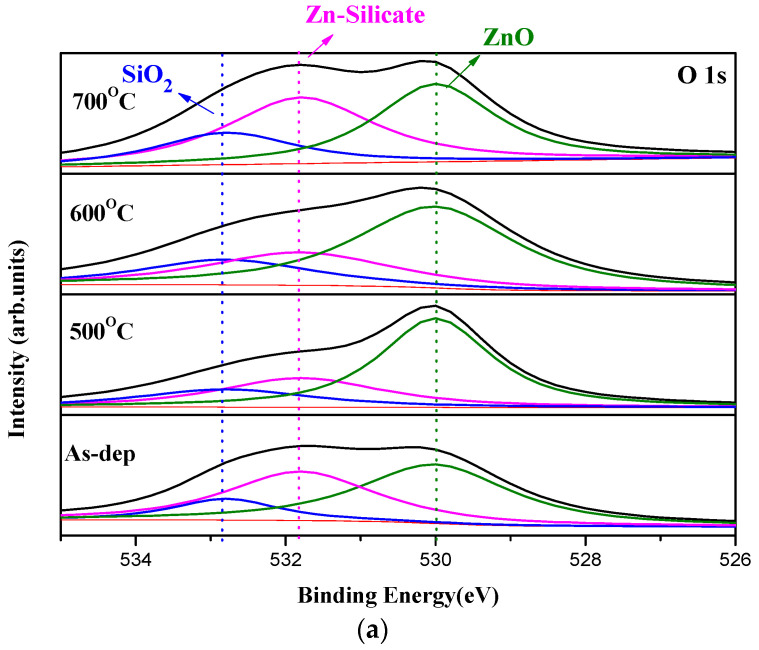
The O1s XPS spectra of (**a**) ZnO, (**b**) AlZnO, (**c**) Al_2_O_3_, (**d**) Al_2_O_3_-doped ZnO sensing films annealed at various temperatures in O2 ambient.

**Figure 4 membranes-12-00168-f004:**
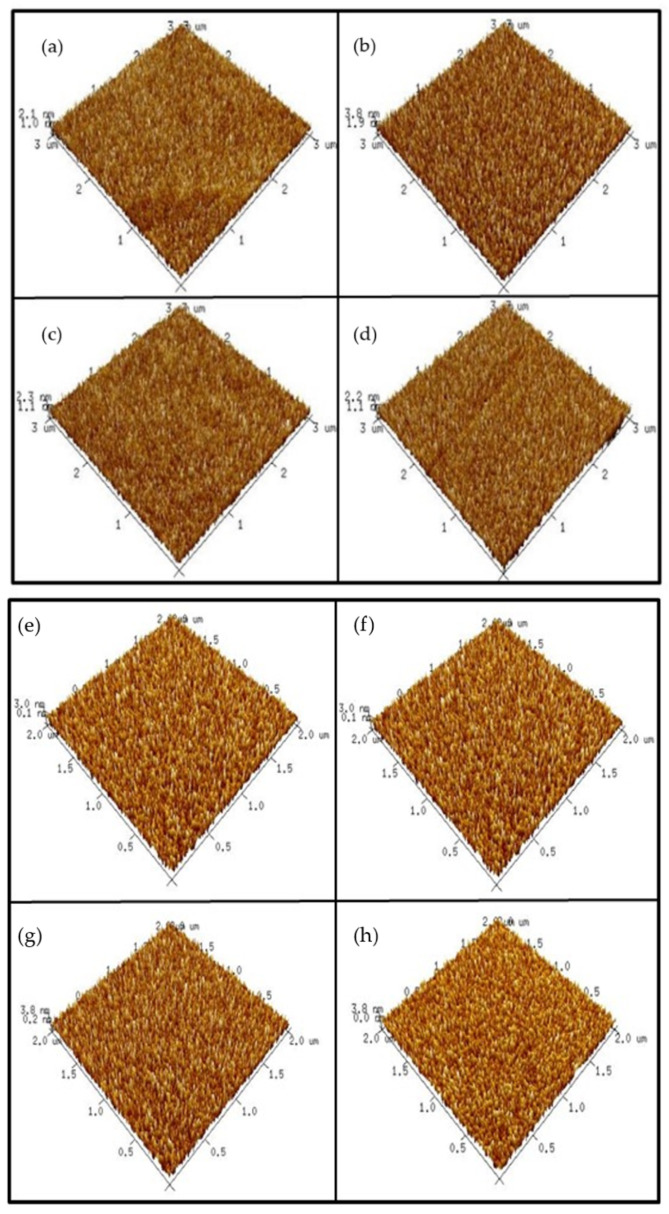
AFM images of ZnO film (**a**) without annealing and with annealing at (**b**) 500 °C, (**c**) 600 °C, (**d**) 700 °C; AlZnO film (**e**) without annealing and with annealing at (**f**) 500 °C, (**g**) 600 °C, (**h**) 700 °C; Al_2_O_3_ film (**i**) without annealing and with annealing at (**j**) 500 °C, (**k**) 600 °C, (**l**) 700 °C; Al_2_O_3_-doped ZnO film (**m**) without annealing and with annealing at (**n**) 500 °C, (**o**) 600 °C, (**p**) 700 °C.

**Figure 5 membranes-12-00168-f005:**
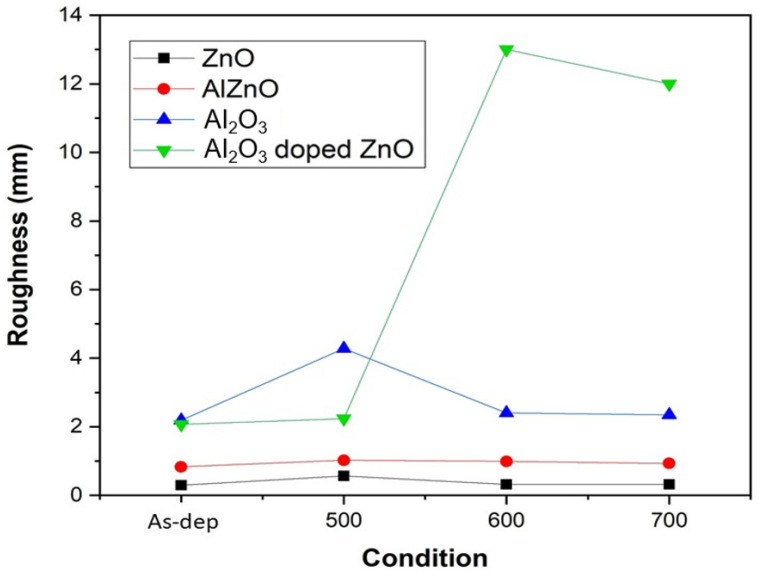
The comparison graph of various samples annealed at different RTA temperatures.

**Figure 6 membranes-12-00168-f006:**
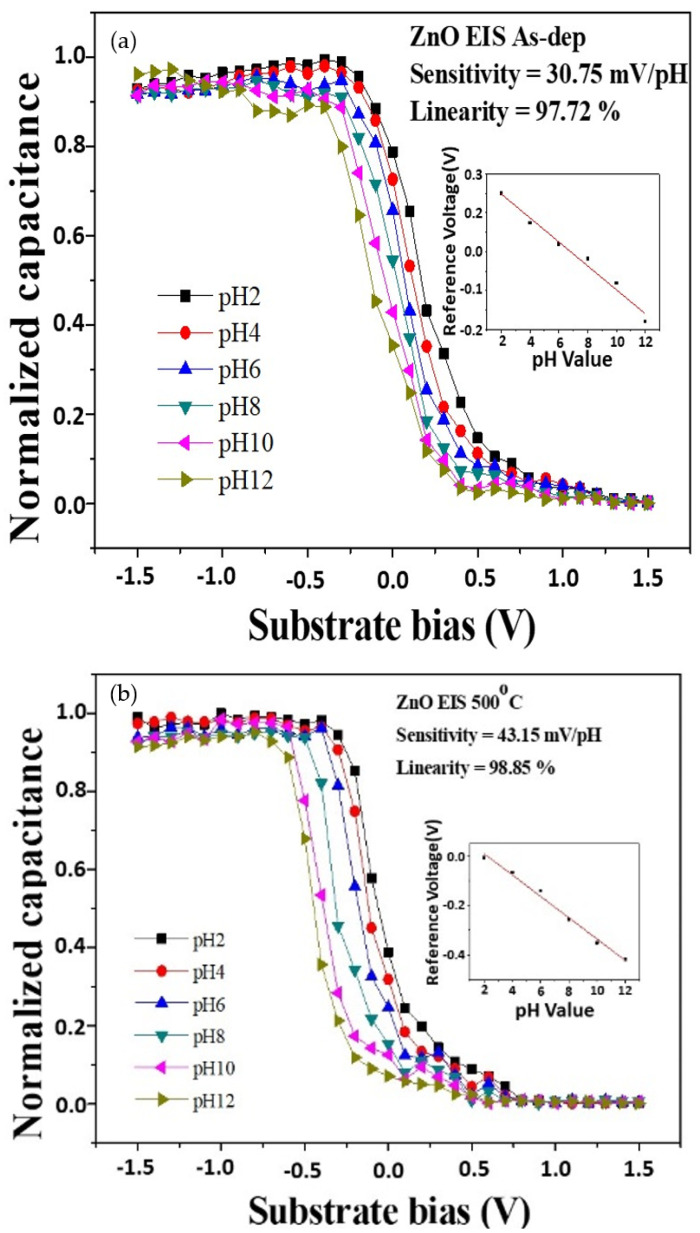
The normalized C–V curve of (**a**) the as-deposited ZnO sensing membrane, (**b**) the ZnO sensing membrane annealed at 500 °C in O_2_, (**c**) the as-deposited AlZnO sensing membrane, (**d**) the AlZnO sensing membrane annealed at 500 °C in O_2_, (**e**) the as-deposited Al_2_O_3_ sensing membrane (**f**) the Al_2_O_3_ sensing membrane annealed at 500 °C in O_2_, (**g**) the as-deposited Al_2_O_3_-doped ZnO sensing membrane (**h**) the Al_2_O_3_-doped ZnO sensing membrane annealed at 500 °C in O_2_. All the inset figures represent the sensitivity and linearity calculations.

**Figure 7 membranes-12-00168-f007:**
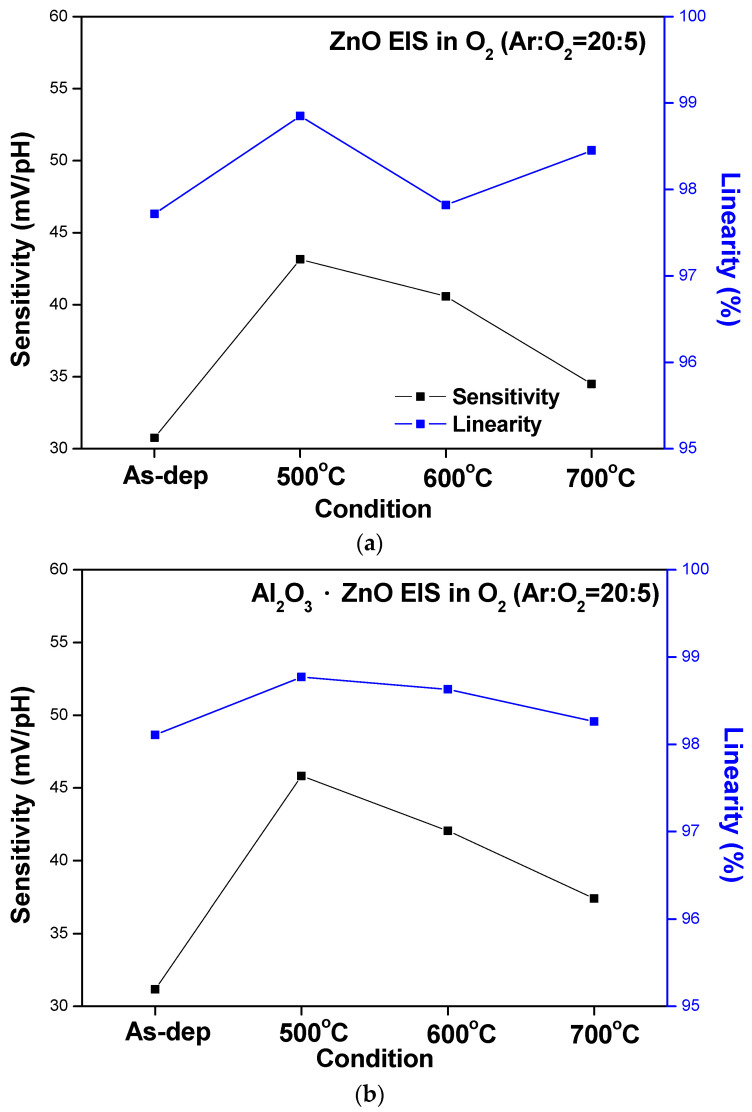
The sensitivity and linearity of (**a**) ZnO, (**b**) AlZnO, (**c**) Al_2_O_3_, (**d**) Al_2_O_3_-doped ZnO sensing membranes with different RTA temperatures in O_2_ ambient.

**Figure 8 membranes-12-00168-f008:**
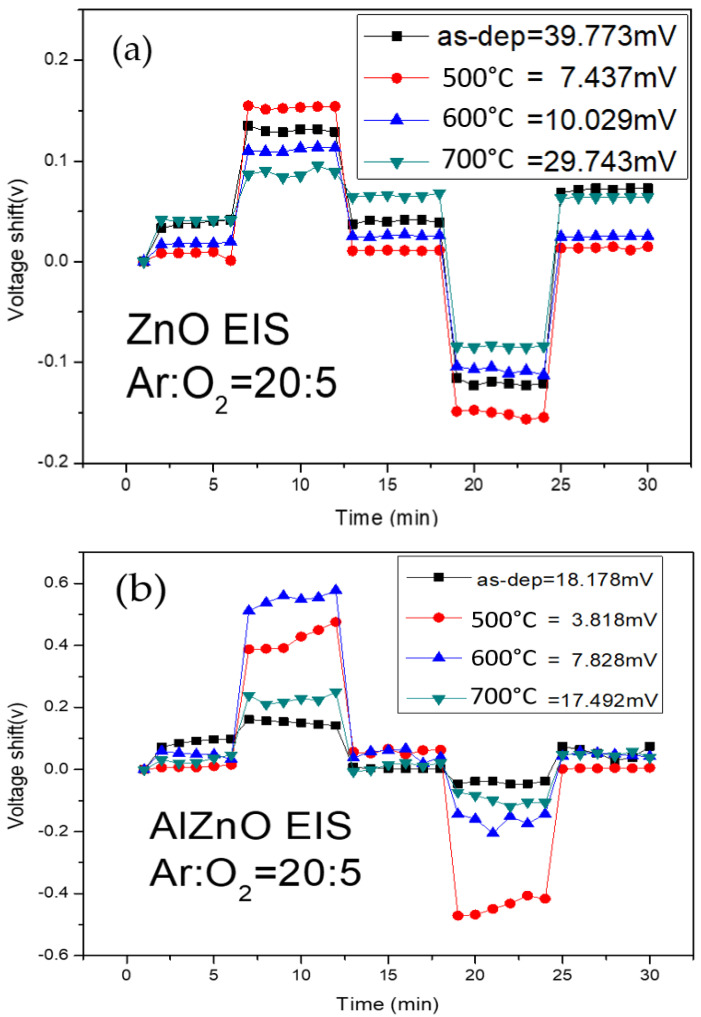
The hysteresis of (**a**) ZnO, (**b**) AlZnO, (**c**) Al_2_O_3_, and (**d**) Al_2_O_3_-doped ZnO sensing membrane with various RTA temperatures in O_2_ ambient during the pH loop of 7→4→7→10→7 over a period of 30 min.

**Figure 9 membranes-12-00168-f009:**
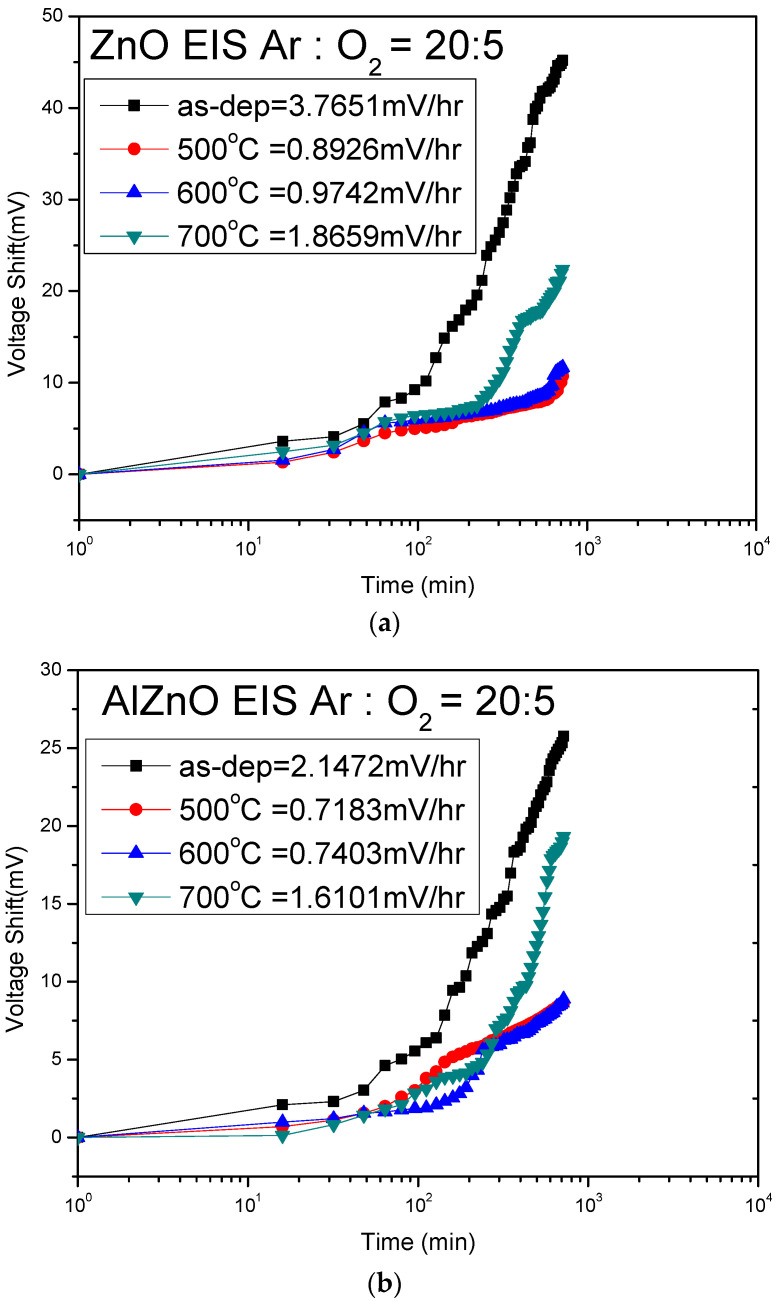
Drift voltage of (**a**) ZnO, (**b**) AlZnO, (**c**) Al_2_O_3_, (**d**) Al_2_O_3_-doped ZnO sensing membranes annealed with various RTA temperatures in O_2_ ambient, then dipped in pH 7 buffer solution for 12 h.

## Data Availability

.Not applicable.

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
