# Peer review of "Comparison of ZnO, Al2O3, AlZnO, and Al2O3-Doped ZnO Sensing Membrane Applied in Electrolyte-Insulator-Semiconductor Structures"

_membranes, 2022, doi:10.3390/membranes12020168_

Round 1
Reviewer 1 Report
I would suggest to read carefully the whole text there are syntax and typos
figure 1a and figure 1b is similar structure, aside from membrane?
How suppressing formation of Zn silicate improve sensing performance, need elaborate more.
What material quality from XRD observation and XPs that the authors said is improved especially at 500C? More crystalline or etc?(line 118)
line 131, from where the authors get this assumption?
line 254 how do you proved that, or is that a mere statement from other reference?
Which has a higher influence morphology, crystalline or Al content on the sensitivity?
Reviewer 2 Report
This paper is a study of the Comparison of ZnO, Al2O3, AlZnO, and Al2O3 doped ZnO Sensing Membrane Applied in Electrolyte-Insulator- Semiconductor Structures.
The paper is not well written, and the English is poor.
1. In the “Introduction” part, it is not well written “Proposed in literatures literature, the strategy to integrate”, write ok literature.
2. In the introduction part: „ Combinations of Al2O3 and ZnO to form multifunctional EIS membranes are „, please rephrase Combining ….
3. In chapter 3. Results and Discussion: Figure 2 - please indicate the best result/conclusion.
4. In chapter 3. Results and Discussion: Figure 4. AFM images of ZnO film: please indicate an Rhms of film deposition.
5. In chapter 3. Results and Discussion: Figure 6. “The normalized C-V curve of (a) the as-deposited ZnO sensing membrane, (b)the ZnO 205 sensing membrane annealed at 500°C in O2, (c) the as-deposited AlZnO sensing membrane, (d) the 206 AlZnO sensing membrane annealed at 500°C in O2, (e) the as-deposited Al2O3 sensing membrane 207 (f) the Al2O3 sensing membrane annealed at 500°C in O2, (g) the as-deposited Al2O3 doped ZnO 208 sensing membrane (h) the Al2O3 doped ZnO sensing membrane annealed at 500°C in O2. (All the 209 inset figure represents the sensitivity and linearity calculation.)”, please indicate the best result of the C-V measurement.
5. In chapter 3. Results and Discussion: Figure 8. “The hysteresis of (a) ZnO(b) AlZnO(c) Al2O3 and (d) Al2O3 doped ZnO sensing mem- 264 brane with various RTA temperatures in O2 ambient during the pH loop of 7→4→7→10→7 over a 265 period of 30 minutes.”, please indicate the best result of your hysteresis measurement.
6. In chapter 3. Results and Discussion: Figure 9. “Drift voltage of (a) ZnO(b) AlZnO(c) Al2O3 (d) Al2O3 doped ZnO sensing membrane an- 294 nealed with various RTA temperatures in O2 ambient, then dipped in pH 7 buffer solution for 12 295 hours”, please explain the logarithmic scale of time, and indicate the best result.
7. Please rephrase the conclusions.
I recommend major revision.
Round 2
Reviewer 2 Report
The revised manuscript fulfils all the reviewer requirements.
I recommend being accepted as it is.